# RNAi-Mediated Knockdown of Transcription Factor E93 in Nymphs of the Desert Locust (*Schistocerca gregaria*) Inhibits Adult Morphogenesis and Results in Supernumerary Juvenile Stages

**DOI:** 10.3390/ijms21207518

**Published:** 2020-10-12

**Authors:** Marijke Gijbels, Elisabeth Marchal, Thomas Wolf Verdonckt, Evert Bruyninckx, Jozef Vanden Broeck

**Affiliations:** 1Research group of Molecular Developmental Physiology and Signal Transduction, KU Leuven, Zoological Institute, Naamsestraat 59 box 2465, 3000 Leuven, Belgium; Marijke.Gijbels@kuleuven.be (M.G.); Elisabeth.Marchal@imec.be (E.M.); ThomasWolf.Verdonckt@kuleuven.be (T.W.V.); Evert.Bruyninckx@kuleuven.be (E.B.); 2Life Science Technologies, Imec, Kapeldreef 75, B-3001 Leuven, Belgium

**Keywords:** ecdysteroid, hemimetabola, insect, juvenile hormone, metamorphosis, RNA interference

## Abstract

Postembryonic development of insects is coordinated by juvenile hormone (JH) together with ecdysteroids. Whereas the JH early response gene *krüppel-homolog 1* (*Kr-h1*) plays a crucial role in the maintenance of juvenile characteristics during consecutive larval stages, the ecdysteroid-inducible early gene *E93* appears to be a key factor promoting metamorphosis and adult morphogenesis. Here, we report on the developmental and molecular consequences of an RNAi-mediated knockdown of *SgE93* in the desert locust, *Schistocerca gregaria*, a hemimetabolan species. Our experimental data show that injection of gregarious locust nymphs with a double-stranded RNA construct targeting the *SgE93* transcript inhibited the process of metamorphosis and instead led to supernumerary nymphal stages. These supernumerary nymphal instars still displayed juvenile morphological features, such as a nymphal color scheme and body shape, while they reached the physical body size of the adult locusts, or even surpassed it after the next supernumerary molt. Interestingly, when compared to control locusts, the total duration of the fifth and normally final nymphal (N5) stage was shorter than normal. This appeared to correspond with temporal and quantitative changes in hemolymph ecdysteroid levels, as well as with altered expression of the rate-limiting Halloween gene, *Spook* (*SgSpo*). In addition, the levels of the *ecdysone receptor* (*SgEcR*) and *retinoïd X receptor* (*SgRXR*) transcripts were altered, indicating that silencing *Sg*E93 affects both ecdysteroid synthesis and signaling. Upon knockdown of *Sg*E93, a very potent upregulation of the *SgKr-h1* transcript levels was observed in both head and fat body, while no significant changes were detected in the transcript levels of *SgJHAMT* and *SgCYP15A1*, the enzymes that catalyze the two final steps in JH biosynthesis. Moreover, the process of molting was disturbed in these supernumerary nymphs. While attempting ecdysis to the next stage, 50% of the N6 and all N7 nymphal instars eventually died. *S. gregaria* is a very harmful, swarm-forming pest species that destroys crops and threatens food security in many of the world’s poorest countries. We believe that a better knowledge of the mechanisms of postembryonic development may contribute to the discovery of novel, more selective and sustainable strategies for controlling gregarious locust populations. In this context, identification of molecular target candidates that are capable of significantly reducing the fitness of this devastating swarming pest will be of crucial importance.

## 1. Introduction

Juvenile hormone (JH) together with ecdysteroids plays a critical role in insect postembryonic development. Where the molting process in juvenile insects is triggered by periodic pulses of ecdysteroids, the absence or presence of JH determines the nature of the molt [1]. The ecdysteroid signal is mediated by binding to a heterodimeric complex of two nuclear receptors (i.e., the ecdysone receptor (EcR) and the retinoid-X-receptor/ultraspiracle (RXR/USP)), as reviewed by Hill et al. (2013) [2]. JH is a sesquiterpenoid hormone synthesized in the *corpora allata* (CA) of insects. The molecular signaling pathway induced by JH has only recently been discovered with the characterization of a JH receptor, methoprene-tolerant (Met), which is a transcription factor belonging to the basic-helix-loop-helix (bHLH)/Per-Arnt-Sim (PAS) family [3,4]. Binding of JH to Met stimulates the formation of a complex with other bHLH-PAS proteins, such as the steroid receptor co-activator Taiman (Tai) [3]. The resulting JH-Met-Tai complex induces the expression of JH response genes, including the gene coding for *Krüppel-homolog 1* (*Kr-h1*), a Zn-finger containing transcription factor [5,6]. RNA interference (RNAi) studies have demonstrated the importance of the JH receptor Met, as well as the downstream-acting transcription factor Kr-h1, in mediating the anti-metamorphic effects of JH in the red flour beetle *Tribolium castaneum*, the German cockroach *Blattella germanica*, the kissing bug *Rhodnius prolixus*, the two-spotted cricket *Gryllus bimaculatus*, the brown planthopper *Nilaparvata lugens*, the common bed bug *Cimex lectularius*, the migratory locust *Locusta migratoria* and the fire bug *Pyrrhocoris apterus* [7,8,9,10,11,12,13,14,15,16,17].

Our current study focuses on the role of the ecdysteroid-induced early response gene, E93, a helix-turn-helix (HTH) transcription factor. This gene was first reported to be involved in the regulation of programmed cell death in *Drosophila melanogaster* prepupae [18,19]. The concept that E93 is crucial for adult morphogenesis was proposed several years later [20]. Whereas Kr-h1 mediates the *status quo* action of JH, E93 acts as a key determinant that promotes adult morphogenesis. RNAi-mediated knockdown of *E93* in larvae of the holometabolan insect *T. castaneum* has induced reiteration of larval development or resulted in the formation of supernumerary second pupae, depending on the time of dsRNA injection [15,21]. In nymphs of the hemimetabolan insect species *B. germanica*, *G. bimaculatus*, *C. lectularius* and *N. lugens*, knockdown of E93 prevented the nymphal-to-adult transition, giving rise to supernumerary nymphal instars [8,10,12,21]. The gene regulatory mechanism by which JH inhibits metamorphosis involves the transcription factors Met, Kr-h1 and E93, and is defined as the MEKRE93 axis [22]. During juvenile-to-juvenile transitions, high-JH titres induce (via Met) expression of Kr-h1, which in turn represses the expression of *E93*. In the final juvenile stage, *Kr-h1* expression is interrupted by the drop in JH titre, resulting in a strong induction of E93 through ecdysone signaling, which triggers metamorphosis and adult development (as reviewed by Belles, 2019) [23]. An inhibitory effect of Kr-h1 on *E93* expression and *vice versa* has been established in several insect species [15,21,24].

In the present study, we investigate the role of the adult specifier gene *E93* in the postembryonic development of the desert locust, *Schistocerca gregaria*, in its gregarious phase. We describe a supernumerary nymphal phenotype, the result of a redefined development caused by the RNAi-mediated silencing of *E93* starting in the penultimate nymphal stage. Supernumerary nymphs are characterized by a nymphal color scheme while growing to the physical size of an adult locust, as well as the presence of wings, although these are not functional. Moreover, these supernumerary nymphs remain in the fifth nymphal stage for a shorter time when compared to control fifth-nymphal locusts. In this context, we analyze and compare the expression of genes contributing to the MEKRE93 axis and to hormone biosynthetic pathways, as well as the temporal ecdysteroid profiles in the hemolymph of both control and *dsSgE93*-injected nymphal locusts.

## 2. Results

### 2.1. Identification and Phylogenetic Analysis of *SgE93*

The nucleotide and amino acid sequences of *S. gregaria E93* (*SgE93*) were found in the NCBI (accession numbers LC215000.1 and BBA65756.1, respectively [25]) (Appendix A). An identical *SgE93* sequence was also found in the recently published first draft assembly of the *S. gregaria* transcriptome and genome (transcript SCHGR_00000941 from locus 1287777-1299780 of the seq2 contig [26]). The comparison with other E93 sequences revealed that *SgE93* possesses two helix-turn-helix (HTH) DNA binding motifs of the pipsqueak family, named RHF1 and RHF2, one nuclear receptor interaction motif (NR-box, also named the LXXLL motif) and the co-repressor C-terminal-binding protein interaction motif (CtBP-im, also named as PLDLSAK motif) (Appendix A). Our in silico analysis revealed *Bombyx mori* transcripts (as sequenced by read ICPK01035599.1 and ICPK01035595.1) that encode isoforms with high similarity to the predicted *Bm*E93 protein sequences available in the NCBI. The isoform identified from read ICPK01035599.1 was added to the list of orthologs used in the phylogenetic analysis (Appendix A). A maximum-likelihood phylogenetic analysis of E93 sequences for selected species showed that *SgE93* is most closely related to the (partial) orthologous sequence of *G. bimaculatus*, another orthopteran species, followed by a larger group of Blattodea orthologs (Appendix A). 

### 2.2. Knockdown of *SgE93* Resulted in Supernumerary Nymphal Stages

Injections of *dsSgE93* (knockdown condition) or *dsGFP* (control) started on day 0 in the fourth nymphal stage (N4 D0) and were repeated on the day of molting to the following stage(s) (Figure 1A,B). When compared to the *dsGFP*-injected control condition, *Sg*E93 knockdown significantly affected development of *S. gregaria* (Figure 1C–E and Figure 2). All of the experimental (*dsSgE93*) locusts molted into a supernumerary (i.e., sixth) nymphal stage (N6), while all control (*dsGFP*) locusts successfully completed their final molt to the adult stage (Figure 1C–E and Figure 2B). The supernumerary nymphs displayed the typical color pattern of a juvenile locust, whereas they had a physical body size similar to (in the case of N6) or even larger than (in the case of N7) that of an adult locust. Moreover, non-functional wings were visible, which were larger than the wing primordia of normal N5 nymphs, but had a smaller, deformed and less pigmented appearance when compared to those of adult locusts (Figure 1C–E). Figure 1D,E show pictures of the different developmental stages of *S. gregaria*, including the supernumerary sixth (N6) and seventh (N7) nymphal stages that were obtained after *Sg*E93 knockdown during this study. Although the supernumerary stages clearly display the typical nymphal characteristics, a very notable difference can be observed in body size and wings when compared to normal fifth-nymphal (N5) locusts (*dsGFP* control).

### 2.3. *SgE93* Silencing Altered the Duration of the Molting Cycle and Affected the Success of Molting to Later Nymphal Stages 

*dsSgE93-*injected locusts remained in the fourth nymphal stage (N4) for a significantly shorter period when compared to the control (*dsGFP*-injected) locusts, with a median duration of five and six days, respectively (Figure 2A). The knockdown (*dsSgE93-injected*) animals also remained in the fifth nymphal stage for a significantly shorter period than the control (*dsGFP*-injected) locusts (Figure 2B). For the locusts of the *Sg*E93 knockdown condition, which molted to the supernumerary nymphal N6 stage, the median duration of the N5 stage was only five days, resembling the normal duration of the N4 stage, while for those molting (and metamorphosing) to the adult stage (*dsGFP*) this was nine days (Figure 2A,B). Moreover, the supernumerary N6 nymphal insects further developed to generate a supernumerary N7 stage (Figure 2C,D). However, six out of 12 locusts did not successfully complete their molt to this seventh nymphal stage (N7). After initiating ecdysis, the respective locusts appeared to be unable to shed their old cuticle and eventually died (Figure 2D). None of the supernumerary N7 nymphal locusts survived the molting process to the next nymphal stage (N8). The supernumerary nymphs that died during ecdysis to N7 or N8 all displayed a juvenile coloration pattern and had short, deformed wings. Moreover, adult genitalia were not observed.

### 2.4. Transcript Levels of MEKRE93 and JH Biosynthesis Pathway Components in *dsSgE93*- and *dsGFP*-Injected Locusts 

The levels of transcript coding for the components of the MEKRE93 pathway, as well as the enzymes catalyzing the two final steps of JH biosynthesis, were determined in head and fat body samples in fifth-nymphal (N5) locusts on day four after the N4-to-N5 nymphal molt. A significant downregulation of *SgE93* mRNA levels (−54%) was detected in the head of fifth nymphal *dsSgE93*-injected locusts when compared with control (*dsGFP-injected*) locusts, while no significant difference was observed in the fat body at this time point (Figure 3C). In comparison with the control (*dsGFP*) locusts, *SgKr-h1* mRNA levels were significantly higher in both head and fat body of *dsSgE93*-injected locusts on day four of the fifth nymphal stage (N5 D4) (Figure 3B). For *SgMet*, as well as the two JH biosynthetic enzymes, no significant differences were observed between *dsSgE93*-injected and control (*dsGFP*) locusts (Figure 3A,D,E).

### 2.5. Transcript Levels of Different *Halloween* Genes and Components of the Ecdysone Receptor Complex in *dsSgE93*- and *dsGFP*-Injected Locusts

The expression profiles of different *Halloween* genes, *Spook* (*SgSpo*), *Phantom* (*SgPhm*), *Disembodied* (*SgDib*) and *Shadow* (*SgSad*), which encode Cytochrome P450 enzymes involved in ecdysteroid biosynthesis [27,28], were compared between *dsSgE93-* and *dsGFP-* (control) injected fifth-nymphal (N5) locusts on day four after the N4-to-N5 molt (Figure 4). Relative *SgSpo* mRNA levels of *dsSgE93*-injected locusts were significantly higher in the head, whereas for the other *Halloween* genes no significant differences were observed (Figure 4A–D). In addition, we have also analyzed the transcript levels of *SgEcR* and *SgRXR*, the genes that code for the components of the ecdysone receptor complex in both *Sg*E93 knockdown and *dsGFP* control animals. When compared to control (*dsGFP*) fifth-nymphal locusts, a significant upregulation of *SgRXR* transcript levels was observed in the head of *dsSgE93*-injected nymphs (Figure 4F), while in fat body of these animals, expression of both *SgRXR* and *SgEcR* was significantly upregulated (Figure 4E,F). 

### 2.6. Temporal Profiles of Ecdysteroids in Hemolymph of *dsSgE93*- and *dsGFP*-Injected Locusts

Ecdysteroid levels were measured in hemolymph samples that were collected every day starting on day one of the fifth nymphal (N5 D1) stage (Figure 5 and Appendix A). When comparing control (*dsGFP*) and *dsSgE93*-injected locusts, a peak of ecdysteroids was witnessed a few days before molting. However, a temporal shift of this peak was noticed. Whereas the ecdysteroid titres in the *dsSgE93*-injected condition already started rising on day two in fifth-nymphal (N5 D2) locusts, they only rose several days later in the control (*dsGFP*) (N5 D5) locusts (Appendix A). Nevertheless, the overall quantity of ecdysteroids measured in *dsSgE93*-injected fifth-nymphal locusts tended to be lower when compared to the control (*dsGFP*) (N5) locusts (Appendix A). Although the duration of the N5 stage was far from uniform for all nymphs within the same condition, the timing of the ecdysteroid peak appeared to be much more synchronized when relating it to the time of molting to the next stage (Figure 5A,B). Also in this comparison, a temporal shift in the ecdysteroid peak was observed, which for the control (*dsGFP*-injected) locusts occurred three days before their final molt to the adult stage and for the locusts of the *SgE93* knockdown condition occurred two days before their molt to a supernumerary N6 stage (Figure 5A,B). Ecdysteroid titres that were measured during the sixth (N6) nymphal stage before molting to a seventh one showed a more variable profile with ecdysteroids peaking two to three days before molting (Figure 5C and Appendix A). Appendix A shows the ecdysteroid levels as measured in the hemolymph samples derived from each individual locust, demonstrating the existing variation.

## 3. Discussion

The desert locust, *S. gregaria*, is a hemimetabolan insect species that develops through five nymphal stages before entering the final adult stage via an incomplete metamorphosis process. During the final molt, functional wings and reproductive organs are formed. In the present study, we describe the developmental and molecular effects of RNAi-mediated silencing of *SgE93*, which was induced by intrahemocoelic injection of dsRNA at the beginning of each nymphal stage of gregarious desert locusts (*S. gregaria*) starting on the day of their molt to the fourth stage (Figure 1A,B). 

### 3.1. *Sg*E93 is the Desert Locust E93 Ortholog

When comparing the cloned *Sg*E93 with other E93 sequences, a similar structural organization was observed (Appendix A). In the silk moth, *B. mori*, RHF1 and RHF2 domains are critical for inducing the expression of a subset of 20E response genes, including EcR and USP. By contrast, the LXXLL and PLDLSAK motifs of E93 inhibit its transcriptional activity. *Bm*E93 can also interact with USP through the NR-box to impair the transcriptional activity of the USP/EcR heterodimer [29]. Based on the highly conserved protein domains observed in all E93 homologs, *Sg*E93 more than likely functions through similar molecular mechanisms. Our in silico analyses indeed indicate that *Sg*E93 is a true ortholog of known insect E93 genes. The phylogenetic tree (Appendix A) comparing several characterized and putative E93 sequences derived from various arthropod species seems to be well in line with the known evolutionary relationships between the represented species and their orders. The obtained tree also overlaps with data published by Chafino et al. (2018) [30], albeit with a clearer grouping of the insect orders. 

### 3.2. *Sg*E93 Knockdown has Inhibited Metamorphosis and Adult Morphogenesis

Our results show that knockdown of *SgE93* interfered with the locusts’ normal development by generating supernumerary nymphal instars instead of reaching the final, adult stage (Figure 1C–E). The appearance of this supernumerary nymphal phenotype indicates that the experimental treatment inhibited the process of incomplete (*cf.* hemimetabolan) metamorphosis. Similar observations of supernumerary nymphal phenotypes were made after silencing E93 in several other insect species, such as *B. germanica*, *G. bimaculatus*, *C. lectularius* and *N. lugens* [8,10,12,21]. Interestingly, for the *dsSgE93*-injected locusts in this study the median duration of the fifth nymphal stage was on average four days shorter than for the *dsGFP*-injected controls, and therefore resembled the normal duration of the N4 stage (Figure 2A,B). This observation indicates that the transition process from N5 to N6 can be considered a true nymphal-to-nymphal transition, by which the locusts remained programmed to invest energy and nutrients in body growth, instead of reallocating their metabolic capabilities to support reproduction and flight in adults. Remarkably, for the *dsSgE93*-injected locusts the median duration of the fourth nymphal stage was also slightly shortened, on average by one day, when compared with the *dsGFP*-injected (control) nymphs (Figure 2A), although this did not affect the nature of the molt. This observation suggests that downregulation of *E93*, even in stages already characterized by low *E93* transcript levels, can still have an influence on developmental timing. Although Ureña et al. (2014) [21] and Li et al. (2018) [11] also started injecting *E93* dsRNA into newly emerged penultimate instar *B. germanica* and *N. lugens*, respectively, they did not check for differences in the duration of this stage. Future experiments during the four initial nymphal stages would enable further investigation of the possible role of *Sg*E93 in developmental timing. The duration of the N6 stage cannot be compared with the *dsGFP*-injected controls, which all reached their final, adult stage. The duration appeared to be more variable, which might refer to the difficulties experienced by many supernumerary nymphs in shedding their old cuticle (Figure 2C,D). Similar difficulties were also reported in E93-silenced supernumerary nymphs of the German cockroach, *B. germanica*, and the brown planthopper, *N. lugens* [12,21]. 

Significant differences in expression levels of the MEKRE93 pathway components *E93* and *Kr-h1* were detected in this study. Normally, the expression level of the adult specifier gene *E93* increases in the second half of the final nymphal stage, when JH levels drop and ecdysteroid levels rise, to ensure the final transition to an adult [23]. Significantly downregulated *SgE93* transcript levels were detected in the head of *dsSgE93*-injected locusts on day four in the fifth nymphal stage, which can explain the transition to a supernumerary nymph instead of an adult locust (Figure 3C). In the fat body, *SgE93* levels were not significantly affected. The exact reason for this is still unclear, but absence of a significant change in *E93* expression was also reported by Ureña et al. (2014) [21], who studied supernumerary nymphs in the German cockroach, *B. germanica*. The observed effects in the fat body on the levels of several other transcripts suggest that *SgE93* has effectively been downregulated in this tissue. Considering the very dynamic nature of its regulation, it seems plausible that *SgE93* gene expression at the examined time point, four days post-injection (N5 D4), may have been influenced in a compensatory manner, but still remained below the threshold needed to induce adult morphogenesis. In future studies, a more complete view on the effects of *Sg*E93 knockdown in supernumerary nymphs would probably be obtained by an analysis of the changes in gene expression of MEKRE93 and ecdysteroid signaling pathway components in a wider range of tissues.

At the same time point (i.e., day four of the fifth nymphal stage (N5 D4)), very significantly upregulated *SgKr-h1* transcript levels were detected in *dsSgE93*-injected locusts in head and fat body samples (Figure 3B). Indeed, *Kr-h1* is an early JH response gene, which has been reported in several insect species to mediate the metamorphosis-inhibiting *status quo* action downstream of the JH signal and its receptor Met [7,9,11,13,31,32]. *Kr-h1* levels normally drop in the final nymphal stage, when JH titers are low, to ensure a final transition to the adult stage [23]. The very significantly increased *SgKr-h1* expression in *dsSgE93*-injected desert locust nymphs may thus have inhibited metamorphosis to the adult stage and instead resulted in supernumerary juvenile stages. In the German cockroach, *B. germanica*, Ureña et al. (2014) [21] demonstrated that the metamorphosis process is not exclusively channeled through a reduction of JH and a consequent decrease of Kr-h1, but that the ecdysone-inducible transcription factor E93 also plays a crucial role in controlling the developmental switch from nymph to adult. Moreover, these authors also showed the existence of an inverse relationship between *Kr-h1* and *E93* expression resulting from mutual repression effects. Our current findings therefore appear to be in agreement with the MEKRE93-based regulatory model for the control of metamorphosis (as reviewed by Belles, 2019) [23]. According to this recently proposed model, low JH titres in the final nymphal stage result in decreased signaling via Met, and this leads to reduced *Kr-h1* expression, which will further result in de-repression of the *E93* gene when a certain threshold is reached. The temporal onset of *E93* expression may be triggered by reaching a critical size and is mediated by ecdysteroid signaling [21,33]. Subsequently, full expression of E93 contributes to the repression of the *Kr-h1* gene. The competence of juvenile stages to undergo metamorphosis and become an adult is neither solely dependent on a low JH titre followed by a decrease in *Kr-h1* expression, nor on an increased ecdysteroid titre followed by an increase in *E93* expression, but also on the crosstalk between different signaling systems, including the regulatory networking interactions between their downstream mediators [21]. In this context, it is interesting to mention the study by Santos et al. (2016) [34], who demonstrated that the expression of *E93* in the final nymphal stage of *B. germanica* is also co-activated by Smox–Medea, two components of the transforming growth factor-β (TGF-β) signaling pathway. In addition, the existence of an epistatic relationship between fushi tarazu factor-1 (FTZ-F1) and E93 was recently proposed in the German cockroach *B. germanica* [35,36,37]. These studies further emphasize the complexity of the gene regulatory network controlling adult morphogenesis.

### 3.3. *Sg*E93 Knockdown has Influenced the Ecdysteroid Pathway

Pulses of ecdysteroids are known to initiate the molting process in insects (as reviewed by Yamanaka et al., 2013) [38]. The Enzyme ImmunoAssay-based quantification of ecdysteroids in this study showed that ecdysteroid levels peaked in hemolymph of *dsGFP*-injected (control) fifth-nymphal locusts three days before the final molt to the adult stage (Figure 5A and Appendix A). In hemolymph of *dsSgE93*-injected fifth-nymphal locusts, a temporal shift in the ecdysteroid profile was witnessed, characterized by an accelerated temporal increase (Appendix A), as well as a reduced time span between the ecdysteroid peak and the next molt (Figure 5B and Appendix A), in comparison with the control (*dsGFP*) N5 locusts (Appendix A and Figure 5A). These observations probably relate to the shorter duration of the N4 and N5 nymphal stages in the *Sg*E93 knockdown condition when compared to the control condition (Figure 2A,B). The interindividual variation of N6 nymphs was much higher (Figure 5C and Appendix A), which may refer to a wider variation in duration of this stage, as well as to the fact that many of these locusts had problems shedding their cuticle (Figure 2C,D). 

In the current study, the significant increase of relative *SgSpo*, *SgEcR* and *SgRXR* transcript levels upon knockdown of *Sg*E93 (Figure 4) might further explain the observed temporal shift in the ecdysteroid profile (Figure 5A,B and Appendix A) and a shorter nymphal-to-nymphal (instead of a final nymphal-to-adult) molting cycle (Figure 1C–E and Figure 2B). On day four in the fifth nymphal stage (N5 D4), ecdysteroid titres still remained low in hemolymph of *dsGFP*-injected (control) locusts, whereas they were significantly increased in hemolymph of *dsSgE93*-injected locusts (Appendix A). The elevated expression levels of the rate-limiting enzyme *SgSpo* [39], measured at this time point in head samples of *dsSgE93*-injected locusts, containing the ecdysteroid biosynthesizing prothoracic glands [40], correlate well with these higher ecdysteroid titres (Figure 4A). The significantly increased *SgEcR* and *SgRXR* transcript levels, in turn, might be responsible for a high responsiveness of the *dsSgE93*-injected locusts to these precociously circulating ecdysteroids, which may further explain the shorter duration of their fifth nymphal stage and, consequently, in combination with high *Sg*Kr-h1 expression levels, the transition to a new nymphal stage instead of metamorphosis. 

Although the ecdysteroid levels detected in hemolymph of *dsSgE93*-injected fifth-nymphal locusts increased faster, they did not reach the same levels as measured in control fifth-nymphal locusts (Figure 5A,B and Appendix A). This observation may be explained by the reorientation of the fifth and normally final metamorphic molt into a supernumerary nymphal molt, since Morgan and Poole (1976) [41] already reported more than 40 years ago that the final (i.e., fifth) nymphal stage of *S. gregaria* was characterized by higher ecdysteroid levels than the penultimate (i.e., fourth) one. A recent study by Lenaerts et al. (2017) [42] corroborated this finding. However, Tawfik et al. (1996) [43] reported the reverse situation, with the fourth nymphal stage of *S. gregaria* having more ecdysteroids than the fifth. Later studies have also suggested that the absolute peak levels of circulating ecdysteroids might be less crucial in initiating molting in *S. gregaria* than the timing of the ecdysteroid surges. Marchal et al. (2011 [27]; 2012 [28]) studied the role of the Halloween genes *Spo*, *Phm* and *Shd* in ecdysteroidogenesis in the desert locust *S. gregaria*. Although circulating ecdysteroid levels were found to be significantly reduced upon RNAi knockdown of the *Halloween* genes, no disruptions of the molting process were observed. Several explanations for this remarkable observation have been suggested that may also apply to the current study. First, it is possible that a sufficient threshold level of 20E needed to complete the molting process was still reached. It must be kept in mind that the L2 polyclonal antibody used for the ecdysteroid measurements exhibits a strong affinity for E, but a six-to-eight times lower affinity to 20E [44]. Another possibility is that perhaps, a portion of the ecdysteroids was hidden from detection, since conjugated forms of ecdysteroids are not detected by this antiserum. Moreover, physiological compensatory mechanisms might also occur, resulting in (for instance) an upregulation and/or activation of ecdysteroid signaling pathway components. Further studies have shown that knockdown of the ecdysteroid receptor complex can indeed severely disrupt the molting process [45]. Interestingly, in the current study, expression of both *SgEcR* and *SgRXR* was increased upon silencing of *Sg*E93, which may have compensated for reduced absolute ecdysteroid peak levels.

In conclusion, our study has clearly shown that experimental manipulation by knockdown of a single molecular target (i.e., the ‘adult specifier’ *Sg*E93) resulted in a severe disruption of post-embryonic development in the desert locust, a very harmful swarming pest species [46,47]. Due to recent political issues, global warming together with poor monitoring has led to a huge plague of *S. gregaria* [48]. Remarkably, other species such as the South American locust (*Schistocerca cancellata*) [49] and the Moroccan locust (*Dociostaurus maroccanus*) [50,51] are currently also causing problems in different parts of the world. The study of crucial processes, such as post-embryonic development and metamorphosis, may reveal novel molecular targets for the discovery of more bio-rational and eco-friendly locust management strategies that cause less side effects than the currently applied insecticides.

## 4. Materials and Methods

### 4.1. Rearing of Animals

The desert locusts (*S. gregaria*) were reared under crowded conditions at a controlled temperature of 32 ± 1 °C, a constant day/night cycle (13 h photophase; 11 h scotophase) and an ambient relative humidity between 40% and 60%. The animals were fed fresh cabbage leaves supplemented with dry oat flakes *ad libitum.* Pots containing a slightly moistened mixture of turf and sand were placed in the breeding cages allowing mated females to deposit their eggs. Once a week, these pots were set apart in clean cages to allow eggs to hatch into first instar locusts.

### 4.2. Tissue Collection

The tissues of interest (head and fat body) were dissected under a binocular microscope and rinsed in *S. gregaria* Ringer solution (1 L: 8.766 g NaCl; 0.188 g CaCl_2_; 0.746 g KCl; 0.407 g MgCl_2_; 0.336 g NaHCO_3_; 30.807 g sucrose; 1.892 g trehalose; pH 7.2), whereafter they were snap-frozen by pooling them in MagNA Lyser Green Bead Tubes (Roche, Mannheim, Germany) that were placed in liquid nitrogen to prevent RNA degradation. For every tissue that was analyzed, samples were collected in five different pools of three dissected animals each and stored at −80 °C until further processing. For analyzing possible effects on expression of MEKRE93 pathway components and enzymes involved in hormone biosynthesis, the head was collected, since this contains both the CA and prothoracic glands [40] that synthesize JH and ecdysteroids, respectively. The fat body, as the main metabolic tissue, was collected to investigate possible effects on expression of MEKRE93 pathway and ecdysteroid receptor components.

### 4.3. RNA Extraction and cDNA Synthesis

Samples of head and fat body, which were obtained from fifth-nymphal (N5) desert locusts on day four (D4) after the N4-to-N5 molt were homogenized in MagNA Lyser Green Bead Tubes (Roche, Mannheim, Germany) using a MagNA Lyser instrument (30s, 6500 rpm; Roche, Mannheim, Germany). Subsequently, total RNA was extracted from these homogenates using the RNeasy Lipid Tissue Kit (Qiagen, Austin, TX, USA) following the manufacturer’s protocol. To prevent genomic DNA contamination, an on-column DNase digestion (RNase-free DNase set, Qiagen, Austin, TX, USA) was performed. The quality and concentration of the resulting RNA samples were measured using a Nanodrop spectrophotometer (Nanodrop ND-1000, Thermo Fisher Scientific, Inc., Waltham, MA, USA). From each RNA sample, cDNA was synthesized by reverse transcription. Of each sample, equal quantities of RNA (500 ng) were reverse-transcribed using a mix of random hexamers and oligo(dT) primers according to the manufacturer’s protocol (PrimeScript RT Reagent Kit, TaKaRa, Invitrogen Life Technologies, Carlsbad, CA, USA). The obtained cDNA was then diluted 10-fold with Milli-Q water (Merck Millipore, Darmstadt, Germany).

### 4.4. Multispecies E93 Gene Identification and Phylogenetic Analysis

The previously published sequence of *SgE93* (LC215000.1, Appendix A [25]) was verified by using it as a query for a BLASTn (version 2.5.0+, expectation value 0.001 [52]) search towards the most recently curated *Schistocerca gregaria* transcriptome and genome data [26]. Less similar sequences were identified using a tBLASTn (version 2.5.0+, expectation value 0.001) search using the protein sequence BBA65756.1 as the query and the most recent *Schistocerca gregaria* transcriptome data as the subject. Published and predicted E93 orthologs from different species were collected from the NCBI (Appendix A). As the published *Bombyx mori* E93 protein with accession number AIL29268.1 was significantly shorter than the orthologous sequences, a tBLASTn search was performed using the same parameters as previously described, using the AIL29268.1 sequence as the query and the NCBI Transcriptome Shotgun Assembly database limited by taxonomy for *Bombyx mori* as the subject. Significant matches were analyzed, the open reading frames identified, and the best candidate was added to the ortholog collection. These E93 sequences were then aligned using MUSCLE (gap penalties: Gap Open −2.9, Gap Extend 0, Hydrophobicity Multiplier 1.2, 20 iterations [53]) and directly used for a maximum likelihood analysis with IQ-tree (version 1.6.12, VT+F+R5 model, 1000 SH-aLRT replicates and 10,000 ultrafast bootstrap replicates [54,55]). Arachnida sequences were used as the outgroup.

### 4.5. Quantitative Real-Time PCR

Quantitative real-time PCR (qRT-PCR) primers for the different target genes are given in Appendix A and Appendix A. These primers were designed using the Primer Express software (Applied Biosystems, Foster City, CA, USA) and primer pairs were validated by analyzing relative standard curves for gene transcripts with serial dilutions (5×) of appropriate cDNA samples. The correlation coefficient (R² = 0.995–1) and efficiency (Eff% = 90–110%) of the qRT-PCR reaction were determined for each primer pair. All qRT-PCR reactions were performed in duplicate in 96-well plates on a StepOne System (ABI Prism, Applied Biosystems, Foster City, CA, USA) following the Fast SYBR Green PCR Master Mix protocol. Each reaction contained 5.0 µL of Fast SYBR Green Master Mix (Applied Biosystems, Foster City, CA, USA), 0.5 µL of each Forward and Reverse primer (10 µM), 1.5 µL of Milli-Q water and 2.5 µL of cDNA. The following thermal cycling profile was applied: 95 °C for 10 min, followed by 40 cycles of 95 °C for 15 s and 60 °C for 60 s, whereafter a melt curve analysis was performed to verify the specificity of the qRT-PCR reactions. A no-template control reaction was always included to confirm absence of contamination, and RT-minus reactions were performed to check for possible genomic DNA contamination. Moreover, the amplification products were analyzed using horizontal agarose gel electrophoresis (1.2% agarose gel containing GelRedTM, Biotium, Fremont, CA, USA) and visualized under UV light. For each transcript, a single band of the expected size was observed, which was further cloned and sequenced (TOPO TA Cloning Kit for sequencing, Invitrogen, Carlsbad, CA, USA) to confirm target specificity.

Optimal housekeeping genes were selected from a pool of candidate housekeeping genes [56] using the geNorm software [57]. In the designed experiment, *EF1α* and *β-actin* appeared to be most stable in the studied head and fat body samples. Relative expression levels were determined according to the comparative Ct method (ΔΔCt) [57]. Statistically significant differences between locusts in the knockdown and control conditions were found via a *t*-test on the log transformed data (with or without a two-sided Welch’s correction) using GraphPad Prism 6 (GraphPad Software Inc., San Diego, CA, USA).

### 4.6. RNA Interference Experiments

Double-stranded (ds)RNA constructs for *SgE93* (*dsSgE93*) (knockdown condition) and *GFP* (*dsGFP*) (control condition) were produced by means of Ambion’s MEGAScript RNAi kit according to the manufacturer’s protocol (Ambion, Life Technologies, Carlsbad, CA, USA; Appendix A). This procedure is based on a high-yield in vitro transcription reaction from a user-provided DNA template with 5′ T7 promoter sequences. Forward and reverse primers flanked by the T7 promoter sequence (given in Appendix A) were used in a PCR reaction with REDTaq DNA polymerase (Sigma-Aldrich, Darmstadt, Germany) to amplify a fragment of the target gene. The resulting amplicons were analyzed using horizontal agarose gel electrophoresis (1.2% agarose gel containing GelRed, Biotium, Fremont, CA, USA) and visualized under UV light. Only a single band was observed, which was subsequently cloned and sequenced (TOPO TA Cloning Kit for sequencing, Invitrogen, Carlsbad, CA, USA) to confirm target specificity. Subsequently, the verified constructs were used in a high-yield in vitro transcription reaction using T7 RNA polymerase. Remaining ssRNA and DNA were removed in a nuclease digestion step. The quality and concentration of the produced dsRNA were determined by means of spectrophotometry (Nanodrop ND-1000, Thermo Fisher Scientific, Inc., Waltham, MA, USA). To confirm dsRNA integrity, a small amount of the reaction product was checked on a 1.2% agarose gel.

Freshly molted fourth-nymphal locusts, synchronized on the day of ecdysis (D0), were each injected with 4 µL *dsSgE93* (500 ng diluted in *S. gregaria* Ringer). Boost injections were given at each molt (i.e., in freshly molted fifth-, sixth- and seventh-nymphal locusts) to ensure a lasting knockdown of *SgE93* mRNA levels. A second group of locusts was injected with *dsGFP* (control) following the same injection scheme until the final molt to adulthood. Tissue samples of control and knockdown conditions were obtained from fifth-nymphal locusts on day four (D4) after the N4-to-N5 nymphal molt. Hemolymph samples were collected every day, starting on day one (D1) after the N4-to-N5 nymphal molt (Figure 1A,B). 

### 4.7. Ecdysteroid Measurement

Using a capillary tube, 5 µL of hemolymph was collected from each animal by piercing the insect’s cuticle behind its hind leg. Subsequently, the collected hemolymph samples were immediately transferred to 95 µL of 100% ethanol and stored at −20 °C until further processing, as previously described by Marchal et al. (2011a) [27]. Ecdysteroid titers were measured using an EIA modified from Porcheron et al. (1989) [58] and discussed by Pascual et al. (1995) [59] and Lafont et al. (2012) [60]. This protocol uses a peroxidase conjugate of 20E as tracer together with rabbit L2 polyclonal antibodies against ecdysteroids. This L2 antiserum has a strong affinity for E, 3-deoxyecdysone and 2-deoxyecdysone, and a six- to eight-fold lower affinity for 20E. Both serum and tracer were very kindly supplied by Prof. J.P. Delbecque (Université de Bordeaux, France). 

### 4.8. Observing Ecdysis

The ecdysis of *dsSgE93-* (knockdown) and *dsGFP-* (control) injected locusts was monitored. Starting from freshly molted fifth-nymphal locusts, the duration and phenotype of each developmental stage were closely observed. Statistically significant differences between *dsSgE93-* and *dsGFP*-injected locusts were found via a log-rank (Mantel–Cox) test using GraphPad Prism 6 (GraphPad Software Inc., San Diego, CA, USA).

## 5. Conclusions

In conclusion, our findings show that the knockdown of *SgE93* in late-nymphal *S. gregaria* resulted in supernumerary nymphal stages characterized by a coloration pattern that is typical for juvenile locusts, the presence of deformed, transparent, non-functional wings and a body size comparable to, or even larger than adult locusts. Moreover, *dsSgE93*-injected fifth-nymphal locusts showed a very prominent increase in *SgKr-h1* relative transcript levels in head and fat body. The duration of the fifth-nymphal stage was significantly shorter in *dsSgE93*-injected animals, compared to the animals in the *dsGFP*-injected control condition. In line with this, a temporal shift was observed in the ecdysteroid profile in hemolymph, as well as altered transcript levels of the *Halloween* gene *SgSpo* and ecdysteroid receptor components *SgEcR* and *SgRXR*. Moreover, some supernumerary N6 and all supernumerary N7 nymphal instars were unable to shed their old cuticles during the molting process and eventually died. Considering the major problems that are still caused by locust swarms in many parts of the world, our study may contribute to future discoveries of novel strategies to control locust population growth and avoid the formation and propagation of devastating swarms.

## Figures and Tables

**Figure 1 ijms-21-07518-f001:**
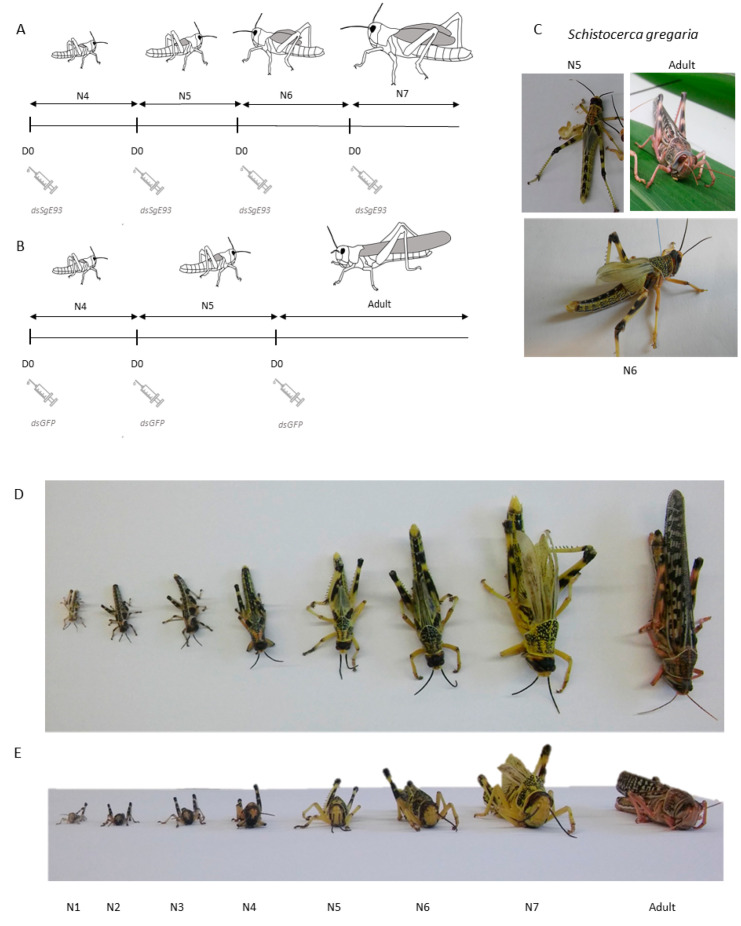
Time scheme of *dsSgE93* injections and phenotype of *dsSgE93*-injected desert locusts. (**A**,**B**) Time scheme showing the different time points of *dsSgE93* (knockdown condition) or *dsGFP* (control condition) injections together with the duration of each developmental stage. Hemolymph samples were collected daily starting on day one in the fifth nymphal stage (N5 D1). (**C**) Observed phenotypes of control (*dsGFP*-injected) N5 (top left) and adult (top right) locusts, as well as a *dsSgE93*-injected (N6; bottom) supernumerary nymphal locust obtained after molting from N5. (**D**,**E**) Representation of postembryonic developmental stages, from hatchling (N1) to adult, including the observed supernumerary nymphal stages (N6 and N7) obtained in this study upon silencing *SgE93*. The normal stages of the desert locust are represented as N1, N2, N3, N4, N5 and Adult. Pictures were taken in (**D**) top view and (**E**) front view.

**Figure 2 ijms-21-07518-f002:**
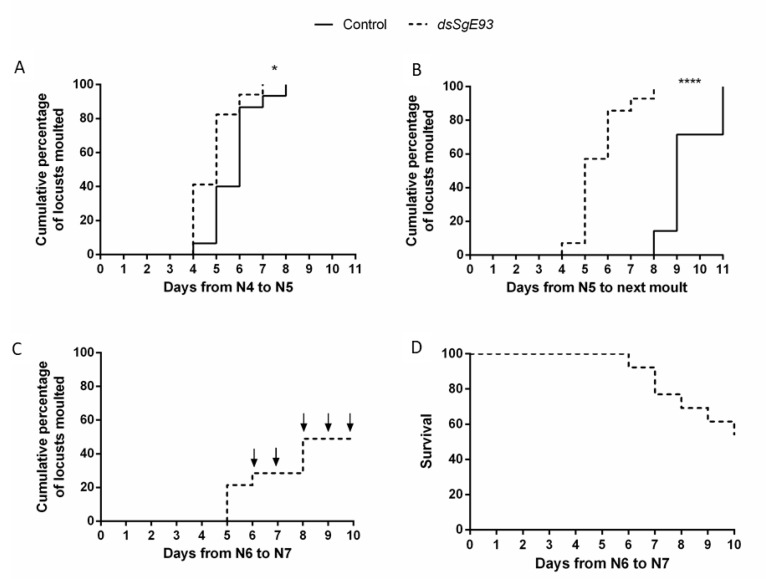
Timing of ecdysis of *dsSgE93*-injected locusts. The timing of ecdysis of *dsGFP*- (control) and *dsSgE93-* (*SgE93* knockdown) injected locusts was observed starting from freshly molted fourth-nymphal locusts. (**A**) The cumulative percentage of 15 *dsGFP-* (control) and 17 *dsSgE93-* (*SgE93* knockdown) injected locusts that molted from the fourth nymphal (N4) stage to the fifth nymphal (N5) stage. (**B**) The cumulative percentage of six *dsGFP-* (control) and 14 *dsSgE93-* (*SgE93* knockdown) injected locusts that molted from the fifth nymphal (N5) stage to the adult or to a sixth nymphal (N6) stage. Statistically significant differences (*p*) between the two conditions were found via a log-rank (Mantel–Cox) test and are indicated by asterisks (* *p* < 0.05; **** *p* < 0.0001). (**C**) The cumulative percentage of 12 *dsSgE93-* (*SgE93* knockdown) injected locusts that successfully molted from the sixth nymphal (N6) stage to the seventh nymphal (N7) stage. Arrows indicate the time points characterized by N6 locusts attempting ecdysis to the next stage, but that eventually died. (**D**) A survival plot representing the percentages of locusts still alive after successfully molting to the seventh nymphal stage (N7).

**Figure 3 ijms-21-07518-f003:**
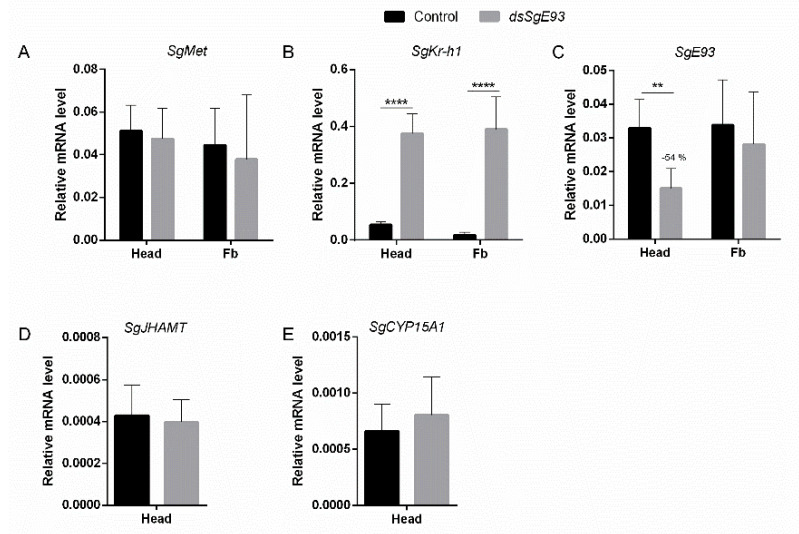
Relative transcript levels of MEKRE93 and JH biosynthesis pathway components in *dsSgE93-* and *dsGFP*-injected desert locust nymphs. Relative (**A**) *SgMet*, (**B**) *SgKr-h1* and (**C**) *SgE93* transcript levels were measured in the fat body and the head of *dsSgE93-* (*Sg*E93 knockdown) and *dsGFP-* (control) injected fifth-nymphal (N5) locusts on day four after the N4-to-N5 molt. (**D**,**E**) Relative *SgJHAMT* and *SgCYP15A1* transcript levels were measured in the head as well. The data represent the mean ± SEM of five independent pools of three animals each, run in duplicate and normalized to *EF1α* and *β-actin* relative transcript levels. Statistically significant differences between the measurements were found via a t-test on log-transformed data (with or without two-sided Welch’s correction) and are indicated by asterisks (** *p* < 0.01; **** *p* < 0.0001).

**Figure 4 ijms-21-07518-f004:**
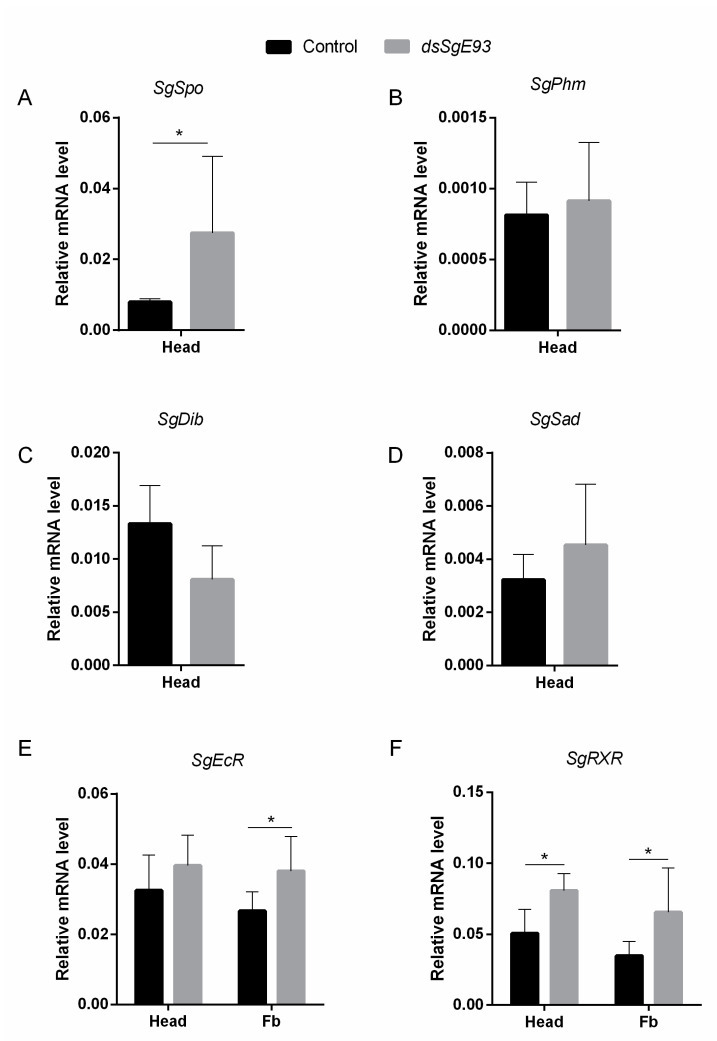
Relative transcript levels of different *Halloween* genes as well as *LmEcR* and *LmRXR* in *dsSgE93-* and *dsGFP*-injected desert locust nymphs. Relative (**A**) *SgSpo*, (**B**) *SgPhm*, (**C**) *SgDib* and (**D**) *SgSad* transcript levels were measured in the head of *dsSgE93-* (*Sg*E93 knockdown) and *dsGFP-* (control) injected fifth-nymphal (N5) locusts on day four after the N4-to-N5 molt. (**E**,**F**) Relative *SgEcR* and *SgRXR* transcript levels were measured in the head and fat body. The data represent the mean ± SEM of five independent pools of three animals each, run in duplicate and normalized to *EF1α* and *β-actin* relative transcript levels. Statistically significant differences between the measurements were found via a *t*-test on log-transformed data (with or without two-sided Welch’s correction) and are indicated by an asterisk (* *p* < 0.05).

**Figure 5 ijms-21-07518-f005:**
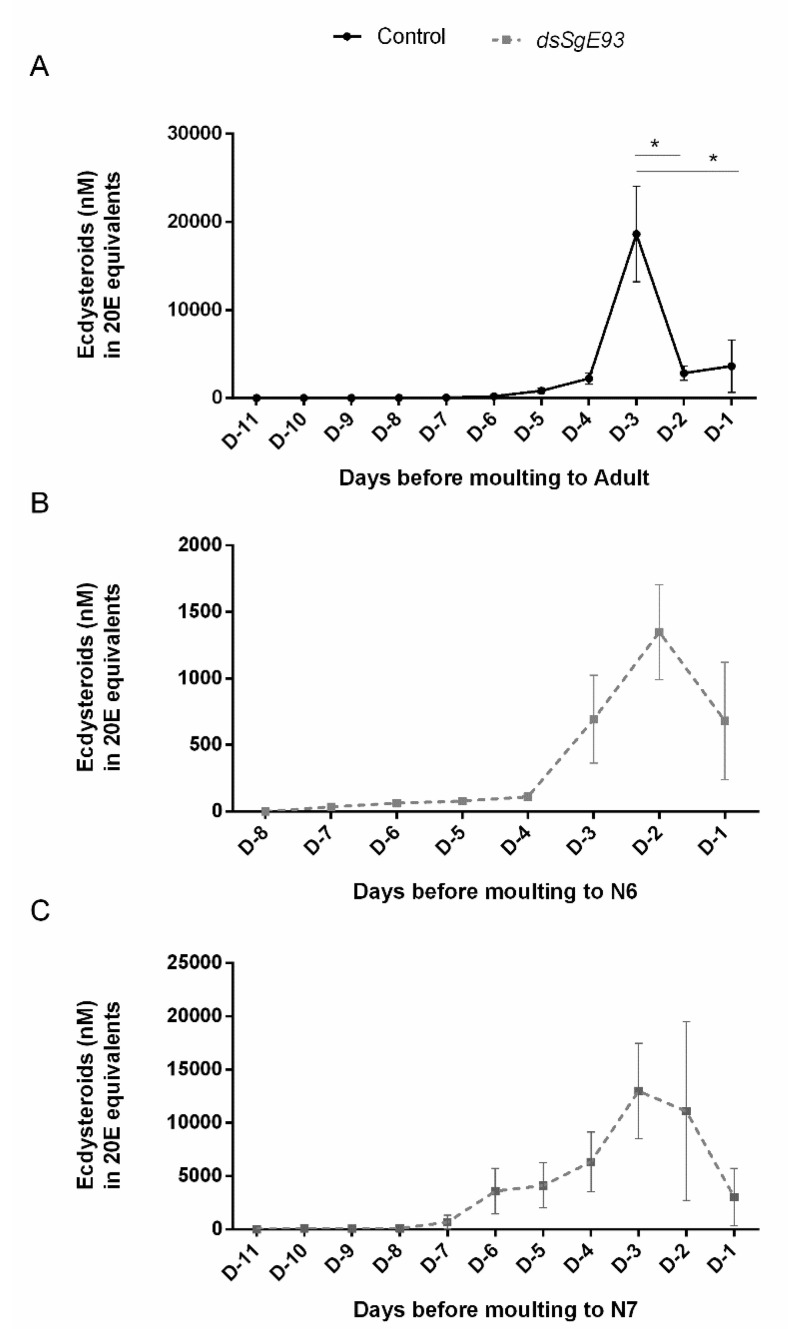
Temporal ecdysteroid profiles in hemolymph of *dsSgE93-* and *dsGFP*-injected desert locust nymphs. Ecdysteroid levels (20E equivalents in nM) were measured in hemolymph samples that were collected daily from *dsGFP*-injected (control) fifth-nymphal (N5) locusts (**A**), as well as *dsSgE93-*injected (*Sg*E93 knockdown) fifth-nymphal (N5) (**B**) and sixth-nymphal (N6) (**C**) locusts. The number of days before molting to the next stage is indicated in the X-axis. Each data point in the figure represents the mean  ±  SEM of five control (*dsGFP*) or six experimental (*dsSgE93*) locusts. Statistically significant differences between the measurements (starting on D-1 until the following ecdysteroid peak) were found via a *t*-test (with or without a two-sided Welch’s correction) and are indicated by an asterisk (* *p* < 0.05).

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
