# Peer review of "RNAi-Mediated Knockdown of Transcription Factor E93 in Nymphs of the Desert Locust (Schistocerca gregaria) Inhibits Adult Morphogenesis and Results in Supernumerary Juvenile Stages"

_ijms, 2020, doi:10.3390/ijms21207518_

Round 1

Reviewer 1 Report

              This paper describes the role of the adult-specifying transcription factor E93 in the locust Schistocerca gregaria using RNAi knockdown as their main tool.  The knockdown based on one injection in the penultimate nymphal instar is about 50%, and they replenish it by injection just after ecdysis to the final nymphal instar (and to the supernumerary stages N6 and N7).  By so doing, they obtain two supernumerary nymphal stages which show no adult wings or characteristic pigmentation but continue to feed and grow.  Some die in ecdysis to N7 and the remainder die during ecdysis to N8 (again with only nymphal characteristics).  Molting to N6 occurs after 5 days as N5 nymphs, whereas the molt to the normal adult occurs after 9 days. The measured ecdysteroid titers confirm this difference in molting times. These E93 RNAi-treated N5 nymphs show an increase in one of the Black Box ecdysone biosynthetic enzymes Spook but no significant changes in any of the other ecdysone biosynthetic enzymes or in the major juvenile hormone (JH) biosynthetic enzymes as measured in the head.  mRNA levels of the two partners of the ecdysone receptor dimer, EcR and RXR, were higher in the fat body, but only RXR was higher in the head. Kr-h1 mRNA levels were higher in the E93 RNAi N5 nymphs in both the head and the fat body, indicating an increased JH titer; but there was no difference in levels of the JH receptor Methoprene-tolerant (Met) mRNA.

            The study is competently done and well controlled, using GFP-RNAi treated nymphs as the controls.  The RNAi suppression seems sufficient based on the findings that E93 RNAi-injected nymphs apparently never become adults. However, a better description of the supernumerary nymphs that die in ecdysis to N7 or N8 is needed.  For instance, is there any sign of adult genitalia or adult cuticle or adult wing development in the pharate N8?  Based on the continued nymphal characteristics seen in the experimental animals, E93 expression must be suppressed in the epidermis, the wings and the genitalia.  These tissues should be assessed for E93 expression in addition to the head and the fat body. In addition, it seems strange that the expression of E93 is only suppressed in the head and not in the fat body.  The explanation given for the lack of effect of the RNAi suppression in the fat body in lines 300-309 seems very weak.  In the nymph, the main function of the fat body is processing incoming nutrients for use in growth by various tissues. Although metamorphosis in hemimetabolous insects usually does not involve a change in diet, there are changes in the fat body, at least in the newly ecdysed adult female, that readies it for its main function of vitellogenin synthesis (see the pioneering papers for Locusta by Wyatt’s lab, reviewed in Wyatt and Davey, Adv. Insect Physiol., 1996). This change from mainly processing nutrients for growth by other tissues to production of a major protein in addition to processing nutrients is definitely a metamorphic change.

              Several other points need attention before publication:

1)  A reorganization of Figures 1 and 2 would be easier for the reader to comprehend.  I suggest putting all the photos in Figure 1 (Fig. 1A, B, C) together with the photos in Figure 2 as a new Figure 1.  Then the new Figure 2 would be the timings shown in the present Figure 1D-G.

2) Figures 3 and 4: One should add a sentence explaining to the reader why you use the head and the fat body as your two sources for mRNA levels.  For looking at effects on the JH receptor and the JH biosynthetic enzymes, the use of the head containing the corpora allata, the source of JH, makes sense.  You should note that the ventral glands (prothoracic gland equivalent) are in the head of the locust rather than in the prothorax as in other insects and give the Joly et al. (Ann. Sci. Natur.Zool. 1956) reference that shows that.

3) Figure 5.  One should show only Figures 5B, D, and F in the text as they are the only relevant data for the timing of the molt.  Figures 5A, C, and E can be put into Figure S5.  It would also be good to measure the ecdysteroid titer in the control N4 molting to N5 as you discuss that in relation to the much lower titer seen in the experimental N5 molting to N6.

4) Figure 4:  It would help the reader to put the gene name in the upper right-hand corner of each of the graphs in this figure in addition to having the name on the X-axis.

5) In the Discussion some of the references are cited as the name, year system and not referred to by number as they are in the Reference list. See lines 57, 280, 300, 321, 327, 337, 341, 354, 390, 392, 412, 418, 543, 544.

6) The Discussion seems unnecessarily long and should be condensed.

7) line 400: ....threshold...

Author Response

We are very grateful to the reviewer for taking the time to read our manuscript and for sharing his/her relevant remarks and constructive suggestions. A point-by-point response to the reviewer’s comments is given below. We have addressed all the listed remarks/questions and revised our manuscript accordingly, as further explained below:

The study is competently done and well controlled, using GFP-RNAi treated nymphs as the controls…

We appreciate this positive feedback and thank the reviewer for the thorough review and detailed comments. The following were listed by the reviewer as comments and minor points to be addressed:

Comment 1: The RNAi suppression seems sufficient based on the findings that E93 RNAi-injected nymphs apparently never become adults. However, a better description of the supernumerary nymphs that die in ecdysis to N7 or N8 is needed. For instance, is there any sign of adult genitalia or adult cuticle or adult wing development in the pharate N8?

Response 1: We absolutely agree with the reviewer and understand that this matter was not sufficiently clear in our first manuscript version. Therefore, we added two more sentences (Revised version lines 182-184) in the Results section, to emphasize that dsSgE93-injected nymphs never became adult: “The supernumerary nymphs that died during ecdysis to N7 or N8 all displayed a juvenile colouration pattern and had short, deformed wings. Moreover, adult genitalia were not observed.”

Comment 2: Based on the continued nymphal characteristics seen in the experimental animals, E93 expression must be suppressed in the epidermis, the wings and the genitalia.  These tissues should be assessed for E93 expression in addition to the head and the fat body. In addition, it seems strange that the expression of E93 is only suppressed in the head and not in the fat body. The explanation given for the lack of effect of the RNAi suppression in the fat body in lines 300-309 seems very weak. In the nymph, the main function of the fat body is processing incoming nutrients for use in growth by various tissues. Although metamorphosis in hemimetabolous insects usually does not involve a change in diet, there are changes in the fat body, at least in the newly ecdysed adult female, that readies it for its main function of vitellogenin synthesis (see the pioneering papers for Locusta by Wyatt’s lab, reviewed in Wyatt and Davey, Adv. Insect Physiol., 1996). This change from mainly processing nutrients for growth by other tissues to production of a major protein in addition to processing nutrients is definitely a metamorphic change.

Response 2: It is a valid suggestion to further assess E93 expression in other tissues, such as the epidermis, the wings and the genitalia, in addition to the head and the fat body. Unfortunately, we did not collect other tissues during this experiment. Crucial endocrine tissues of interest in locusts are prothoracic glands and corpora allata, both situated in the head, which was sampled in this study. In addition, we also included fat body in our study, because this tissue is very important for the control of insect metabolism. Many other, if not all, locust tissues could indeed be of interest too, but of course this would represent an immense extra workload. We added one more sentence (Revised version lines 344-347 in the Discussion section, to emphasize the potential interest of further investigating other tissues as well: “In future studies, a more complete view on the effects of the SgE93 knockdown in supernumerary nymphs would probably be obtained by an analysis of the changes in gene expression of MEKRE93 and ecdysteroid signalling pathway components in a wider range of tissues.”  

We also understand the concern of the reviewer regarding our discussion on the SgE93 transcript levels observed in the fat body. We believe that SgE93 has effectively been downregulated in this tissue, as explained in lines 339-341: “The observed effects in the fat body on the levels of several other transcripts suggest that SgE93 has effectively been downregulated in this tissue.” In all probability, we think that SgE93 gene expression at the examined time point may have been influenced in a compensatory manner as explained in lines 341-344: “Considering the very dynamic nature of its regulation, it seems plausible that SgE93 gene expression at the examined time point, four days post-injection (N5 D4), may have been influenced in a compensatory manner, but still remained below the threshold needed to induce adult morphogenesis.” We also agree with the reviewer that fat body is a crucial tissue for metabolic processes, which are also important during metamorphosis, but we cannot ignore the study by Ureña et al. (2014) [1] either, since also in this study of supernumerary nymphs of the German cockroach, B. germanica, no significant changes in E93 expression were observed in the fat body. Therefore, to be more cautious in discussing this matter, we rephrased this text part (Revised version lines 333-335): “The exact reason for this is still unclear, but an absence of a significant change in E93 expression was also reported by Ureña et al. (2014) [1], who studied supernumerary nymphs in the German cockroach B. germanica.” Moreover, the text part mentioning ‘metamorphic’ and ‘non-metamorphic’ tissues was deleted, since we agree that the interpretation given in our first manuscript version probably was too speculative (Revised version lines 335-339). Moreover, these changes also comply with both reviewers’ request to reduce the length of the Discussion section.

Point 1: A reorganization of Figures 1 and 2 would be easier for the reader to comprehend.  I suggest putting all the photos in Figure 1 (Fig. 1A, B, C) together with the photos in Figure 2 as a new Figure 1.  Then the new Figure 2 would be the timings shown in the present Figure 1D-G.

Response 3: We thank the reviewer for this excellent suggestion. Figures 1 and 2, as well as the corresponding figure legends and references throughout the manuscript, have now been adapted according to the reviewer’s remarks (Revised version lines 112, 114-115, 119, 127-128, 131-148, 151-167, 172, 174, 177-178, 181, 289-290, 302, 321 and 419).

Point 2: Figures 3 and 4: One should add a sentence explaining to the reader why you use the head and the fat body as your two sources for mRNA levels.  For looking at effects on the JH receptor and the JH biosynthetic enzymes, the use of the head containing the corpora allata, the source of JH, makes sense.  You should note that the ventral glands (prothoracic gland equivalent) are in the head of the locust rather than in the prothorax as in other insects and give the Joly et al. (Ann. Sci. Natur.Zool. 1956) reference that shows that.

Response 4: We appreciate the remark and understand that the decision of using head and fat body as the two sources for determining mRNA levels was not sufficiently clarified. Therefore, the following two sentences (Revised version lines 509-513 were added in the Material and Methods section: “For analyzing possible effects on expression of MEKRE93 pathway components and enzymes involved in hormone biosynthesis, the head was collected, since this contains both the CA and prothoracic glands [2], respectively synthesizing JH and ecdysteroids. The fat body, as the main metabolic tissue, was collected to investigate possible effects on expression of MEKRE93 pathway and ecdysteroid receptor components.” We also added the reference suggested by the reviewer.

Point 3: Figure 5. One should show only Figures 5B, D, and F in the text as they are the only relevant data for the timing of the molt. Figures 5A, C, and E can be put into Figure S5. It would also be good to measure the ecdysteroid titer in the control N4 molting to N5 as you discuss that in relation to the much lower titer seen in the experimental N5 molting to N6.

Response 5: We followed the suggestion of the reviewer. Figure 5, as well as the corresponding figure legends, references throughout the manuscript and Supplementary data, have now been adapted as such (Revised version lines 234, 238, 240-241, 243, 246, 249, 256-264, 397, 399-402, 404-406, 418, 422 and 432).

We also agree with the reviewer that it may also have been of interest to know the ecdysteroid titers in the transition from N4 to N5. However, we did not check for this, since our study was primarily focusing on the transition to the supernumerary nymphal stages. Nevertheless, from previous observations of our S. gregaria colony, it is known that N4 locusts are characterized by an earlier ecdysteroid peak when compared to the N5 stage. Moreover, Lenaerts et al. (2017) [3] showed that the ecdysteroid titer detected in N4 locusts is lower than in N5 locusts. We added a short sentence (Revised version line 436) in the Discussion section that refers to these published data: “A recent study by Lenaerts et al. (2017) [3] corroborated this finding.” 

Point 4: Figure 4: It would help the reader to put the gene name in the upper right-hand corner of each of the graphs in this figure in addition to having the name on the X-axis.

Response 6: We thank the reviewer for this comment and followed the suggestion by consistently indicating the gene names on top of each figure panel (and not on either the X- or the Y-axis, while the tissues that were analyzed are now consistently indicated on the X-axis), in both Figures 3 and 4.

Point 5: In the Discussion some of the references are cited as the name, year system and not referred to by number as they are in the Reference list. See lines 57, 280, 300, 321, 327, 337, 341, 354, 390, 392, 412, 418, 543, 544.

Response 7: We thank the reviewer for this keen remark, but we also believe that citing some references in the name-year system (mainly applicable when authors of the cited reference are integrated within the sentence) should not cause any problems if the reference number is added. We understand the confusion and decided to cite the reference number immediately behind the name (year), instead of at the end of the sentence (Revised version lines 51, 78, 285, 311-312, 334, 359, 366, 376, 434, 437 and 440).

Point 6: The Discussion seems unnecessarily long and should be condensed.

Response 8: We agree with the reviewer and decided to omit several redundant sentences throughout the discussion (based on both reviewers’ comments) in order to keep the main focus on the manuscript’s content (Revised version lines 265-493).

Point 7: line 400: ....threshold...

Response 9: We would like to apologize for this mistake. The sentence has been corrected according to the reviewer’s remark (Revised version line 445).

References

[1]      Ureña E, Manjón C, Franch-Marro X, et al. Transcription factor E93 specifies adult metamorphosis in hemimetabolous and holometabolous insects. Proc Natl Acad Sci U S A 2014; 111: 7024–7029.

[2]      Joly P, Joly L, Halbwachs M. Contrôle humoral du développement chez Locusta migratoria. Ann Sci nat Zool 1956; 18: 256–261.

[3]      Lenaerts C, Cools D, Verdonck R, et al. The ecdysis triggering hormone system is essential for successful moulting of a major hemimetabolous pest insect, Schistocerca gregaria. Sci Rep 2017; 7: 46502.

Reviewer 2 Report

Gijbels and his/her colleagues described the functional analyses concerning E93 transcription factor in the desert locust. The findings were supported by the experimental results well. By the way, I have noticed several minor points to finalize this manuscript.

In addition, the following two points should be answered to evaluate correctly. But these comments should not be reflected in the revised version.

1) The error bars in the figures are very long. Are these SEM or SD?? It is not a good impression to understand the variety of the data points. 

2) Very long discussion. If possible, authors should omit several sentences or paragraphs in order to focus this manuscript context.

Minor points)

L15:  is playing -> plays

L133: nymphal -> nymphal stage

Figure 2: If the authors wanted to show the supernumerary phenotype, authors should display the supernumerary nymphal pictures compared with the normal one. Representative one might be fine.

L151 and 153: Time period -> “time” or “period” would be fine.

L273: a juvenile to a juvenile: “a nymph to a nymph”?? I think that this might be re-phrased according to the context.

L294: Clear difference -> Significant difference or Remarkable difference??

Paragraph from L310-L352: Very long paragraph. Please divide several para. Or omit several redundant sentences.

L400: threshold??

Author Response

We are very grateful to the reviewer for taking the time to read our manuscript and for sharing his/her relevant remarks and constructive suggestions. A point-by-point response to the reviewer’s comments is given below. We have addressed all the listed remarks/questions and revised our manuscript accordingly, as further explained below:

… The findings were supported by the experimental results well. By the way, I have noticed several minor points to finalize this manuscript.

We appreciate this positive feedback and thank the reviewer for the thorough review and detailed comments.

Point 1: L15: is playing -> plays

Response 1: The sentence has been corrected according to the reviewer’s remark (Revised version line 15).

Point 2: L133: nymphal -> nymphal stage

Response 2: We understand this remark and added “stage” where we noticed that this word was lacking (Revised version lines 154, 156 and 160).

Point 3: Figure 2: If the authors wanted to show the supernumerary phenotype, authors should display the supernumerary nymphal pictures compared with the normal one. Representative one might be fine.

Response 3: Figure 2 together with panels A – C of Figure 1 now represent Figure 1 according to the other reviewer’s comments. Pictures representing the normal stages of the desert locust (“N1-N5” and “Adult”) are included in this Figure 1D and E. “N6” and “N7” are the supernumerary nymphal stages generated by SgE93 knockdown (Revised version lines 135-136).

Point 4: L151 and 153: Time period -> “time” or “period” would be fine.

Response 4: This has been corrected in our manuscript (Revised version lines 171 and 173).

Point 5: L273: a juvenile to a juvenile: “a nymph to a nymph”?? I think that this might be re-phrased according to the context.

Response 5: We agree with the reviewer and changed the sentence as such (Revised version line 303).

Point 6: L294: Clear difference -> Significant difference or Remarkable difference??

Response 6: This change has been made (Revised version line 327).

Point 7: Paragraph from L310-L352: Very long paragraph. Please divide several para. Or omit several redundant sentences.

Response 7: We agree with the reviewer and decided to omit several redundant sentences. Moreover, we have decided to omit even more sentences throughout the discussion in order to keep the main focus on the manuscript’s content (Revised version lines 265-493).

Point 8: L400: threshold??

Response 8: We would like to apologize for this mistake. The sentence has been corrected according to the reviewer’s remark (Revised version line 445).

In addition, the following two points should be answered to evaluate correctly. But these comments should not be reflected in the revised version.

Comment 1: The error bars in the figures are very long. Are these SEM or SD?? It is not a good impression to understand the variety of the data points.

Response 9: As indicated in the figure legends, error bars represent the SEM. The occurrence of inter-individual variation during post-embryonic development is far from unusual in phytophagous insects, including grasshopper species, as previously reported by Hatle et al. (2003) [1]. In this context, it is very important to emphasize that our conclusions are entirely and exclusively based on observations that are statistically significant. 

Comment 2: Very long discussion. If possible, authors should omit several sentences or paragraphs in order to focus this manuscript context.

Response 10: We agree with the reviewer and also refer to Response 7. 

References:

[1]      Hatle JD, Miller WA, Borst DW. Canalization of development and ecdysteroid timing during the last instar in lubber grasshoppers. J Insect Physiol 2003; 49: 73–80.

Round 2

Reviewer 1 Report

The authors have considerably improved the manuscript by rearranging the figures, adding critical points of information about procedures, and shortening the discussion.  It is now ready for publication.